# Association between maternal vitamin D deficiency and small for gestational age: evidence from a meta-analysis of prospective cohort studies

Yao Chen,[1] Beibei Zhu,[1,2] Xiaoyan Wu,[1,2] Si Li,[3] Fangbiao Tao[1,2]

## ABSTRACT

**Objective** To determine whether maternal vitamin D deficiency during pregnancy is associated with small for gestational age (SGA).

**Methods** A comprehensive literature search of PubMed, the Cochrane Library, Embase, and the Elsevier ScienceDirect library was conducted to identify relevant articles reporting prospective cohort studies in English, with the last report included published in February 2017. Pooled odds ratios (ORs) and corresponding 95% confidence intervals (CIs) were used to evaluate the correlation in a random effects model.

**Results** A total of 13 cohort studies were included in this meta-analysis with a sample of 28 285 individuals from seven countries. The pooled overall OR for babies born SGA was 1.588 (*95%* CI 1.138 to 2.216; p<0.01) for women with vitamin D deficiency. The prevalence of vitamin D deficiency during pregnancy varied from 13.2% to 77.3%. Subgroup analyses identified no significant differences in the association between vitamin D deficiency and SGA based on study quality, gestational week during which blood sampling was performed, cut-off vitamin D levels, sample size, adjustment for critical confounders and method for measuring vitamin D.

**Conclusion** This meta-analysis suggests that vitamin D deficiency is associated with an increased risk of SGA.

[1]Department of Maternal, Child and Adolescent Health, School of Public Health, Anhui Medical University, Hefei, Anhui, China
[2]Anhui Provincial Key Laboratory of Population Health and Aristogenics, Hefei, Anhui, China
[3]Department of Epidemiology and Biostatistics, School of Public Health, Anhui Medical University, Hefei, Anhui, China

**Correspondence to**
Dr Fangbiao Tao;
taofangbiao@126.com

## Strengths and limitations of this study

► To our knowledge, this was the first systematic review that included only prospective cohort studies in its evaluation of the association between vitamin D and small for gestational age (SGA).
► The subgroup analysis performed in this study enabled a more thorough understanding of current evidence.
► Cohort study quality tests, a heterogeneity test, and sensitivity analysis were performed; publication bias was evaluated.
► Different definitions of vitamin D deficiency, insufficiency or sufficiency may have affected the results.
► Substantial heterogeneity existed among several outcomes.

## INTRODUCTION

Vitamin D is fat soluble and a steroid hormone recognised for its major role in calcium metabolism and bone health.[1] Vitamin D deficiency or insufficiency has become a global public health issue,[2] especially for pregnant women, among whom the highest deficiency rate is 84% according to a multiethnic population survey conducted in Norway.[3] Several large-population studies have evaluated the associations of maternal vitamin D deficiency with various adverse maternal and fetal outcomes[4–6] including small for gestational age (SGA).

Infants born SGA are defined as smaller in size than normal for the gestational age, most commonly stipulated by a weight less than the 10th percentile for the corresponding gestational age.[7 8] The incidence of infants who are SGA worldwide is 9.7%,[9] and this percentage is increasing. Infants born SGA have much higher neonatal morbidity and mortality.[10] Katz *et al*[11] demonstrated that the pooled risk ratios (RRs) of neonatal mortality and post-neonatal morbidity in infants who were SGA were 1.83 and 1.90, respectively. SGA may also be strongly correlated with adverse health outcomes in adult life, such as neurocognitive impairment, poor school performance, short stature, and increased risks of diabetes,[12] cardiovascular disease[13] and kidney disease.[14]

Although numerous studies have focused on the association between maternal vitamin D status and SGA, the results of these studies remain inconsistent. A prospective cohort study conducted in the Netherlands evaluated vitamin D concentrations in 3730 pregnant women after 12–14 weeks of gestation and discovered that infants born to mothers with vitamin D deficiency had an increased risk of being SGA compared with those born to mothers with adequate vitamin

D levels.[15] Subsequently, Gernand *et al*[16] reported that if the maternal vitamin D level was less than 15 ng/mL, infants had a significantly higher risk of being SGA. However, other studies have identified no association between vitamin D status and SGA.[17 18]

Given the inconclusive evidence regarding this issue, we summarise the highest quality evidence currently available on the basis of a meta-analysis of prospective cohort studies to determine whether vitamin D deficiency in pregnant women is associated with SGA.

## MATERIALS AND METHODS
### Data sources, search strategy and selection criteria
A systematic literature search was performed using the PubMed, Elsevier ScienceDirect, Cochrane Library, and Embase databases to identify all relevant articles published prior to March 2017. No restrictions were made regarding maternal age and study design. The following keywords were used: 'vitamin D' or 'cholecalciferol' or '25-hydroxyvitamin D' or '25(OH)D' combined with 'SGA' or 'small for gestational age' or 'small-for-gestation-age' or 'small size for gestational age' (see online supplementary box S1 details for the search strategy).

### Selection criteria
We first screened the titles and abstracts of all the articles to identify possible eligible studies and then read the articles in full to determine whether they were in fact eligible. The articles included in the meta-analysis were selected according to the following inclusion criteria: (1) published in English; (2) the population of the study was pregnant women without prechronic disease; (3) only women with singleton gestation were included; (4) the outcome was an infant who was SGA, the control group included women who gave birth to babies not SGA, and the exposure was 'vitamin D deficiency' (25(OH)D<20 ng/mL); (5) study data were in the form of effect estimates (odds ratio (OR) or RR)) and corresponding 95% confidence intervals (CIs), or the article reported data that enable calculation of these; (6) maternal blood samples were taken for assessing 25(OH)D during pregnancy; (7) the study design was that of a cohort study. The final criterion was applied because cohort studies are the most effective means of ascertaining both the incidence and natural history of a disorder. The temporal connection between putative cause and outcome is usually clear in such studies; in addition, the cohort study design reduces the risk of survivor bias. By contrast, this bias often frustrates cross-sectional and case-control studies. For example, case-control studies are more prone to recall and selection biases and are uncertain regarding chronological order, making them of limited use for causal inference.

### Data extraction and quality evaluation
Two investigators reviewed all abstracts of related articles, and read their full text, respectively. We extracted data using a standardised form and assessed study quality. Disagreements were resolved by discussion and consulting a third investigator. The following data were collected from each study: (1) publication information: first author name and publication year; (2) population's characteristics: country of origin, average age and pre-pregnancy body mass index (BMI), ethnicity, education status, current gestational week of blood sampling, gestational age of infant at birth, and season of blood sample; (3) methods: assay of serum or plasma vitamin D levels and sample size; (4) latitude and time of year that data were collected; (5) OR and corresponding 95% CI for each study. If available, ORs with 95% CIs were collected from the original article. If crucial original data were unavailable, ORs with 95% CIs were calculated using other data published in the article to construct 2×2 tables of low vitamin D status versus the presence or absence of SGA. Otherwise, we contacted the corresponding author by e-mail to obtain further details. Finally, we assessed the eligible studies based on the Newcastle Ottawa Scale (NOS). This scale ranges from 0 to 9 and contains nine items (one point for each) in three parts: selection (four items), comparability (two items) and exposure or outcomes (three items). Scores of 0–3 indicated studies to be of poor quality; scores of 4–6 indicated studies to be of moderate quality; and scores of 7 or higher indicated studies to be of high quality (online supplementary box S2).

### Statistical analysis
The data extracted from eligible studies were in the form of effect estimates (OR or RR) and corresponding 95% CIs. Due to the low level of morbidity in babies born of SGA, the OR was approximately equal to the RR.[19] Meta-analysis was performed using the STATA package version 12.0 (Stata Corporation, College Station, Texas, USA). The ORs and 95% CIs for normal vitamin D levels versus deficient vitamin D levels from each study were combined to calculate an estimated pooled OR, 95% CI and p value. The $Q$-statistic test and I-squared ($I^2$) test were used to estimate the heterogeneity among studies.[20] The random effects model is usually more suitable when study data are gathered from the published literature.[21] Therefore, the random effects model was used in our meta-analysis. To evaluate the sources of heterogeneity and the various results obtained for prespecified subgroups, subgroup analysis was performed based on cut-off values, study quality (NOS scores), adjustment for critical confounders, sample size, measurement of vitamin D, and the gestational week in which blood sampling was performed. A sensitivity analysis was conducted to determine the stability and reliability of the results by omitting one study at a time and confirming the consistency of the overall effect estimate. Funnel plots were used to qualitatively assess the publication bias, whereas Egger's and Begg's tests were used to quantitatively assess publication bias.[22 23]

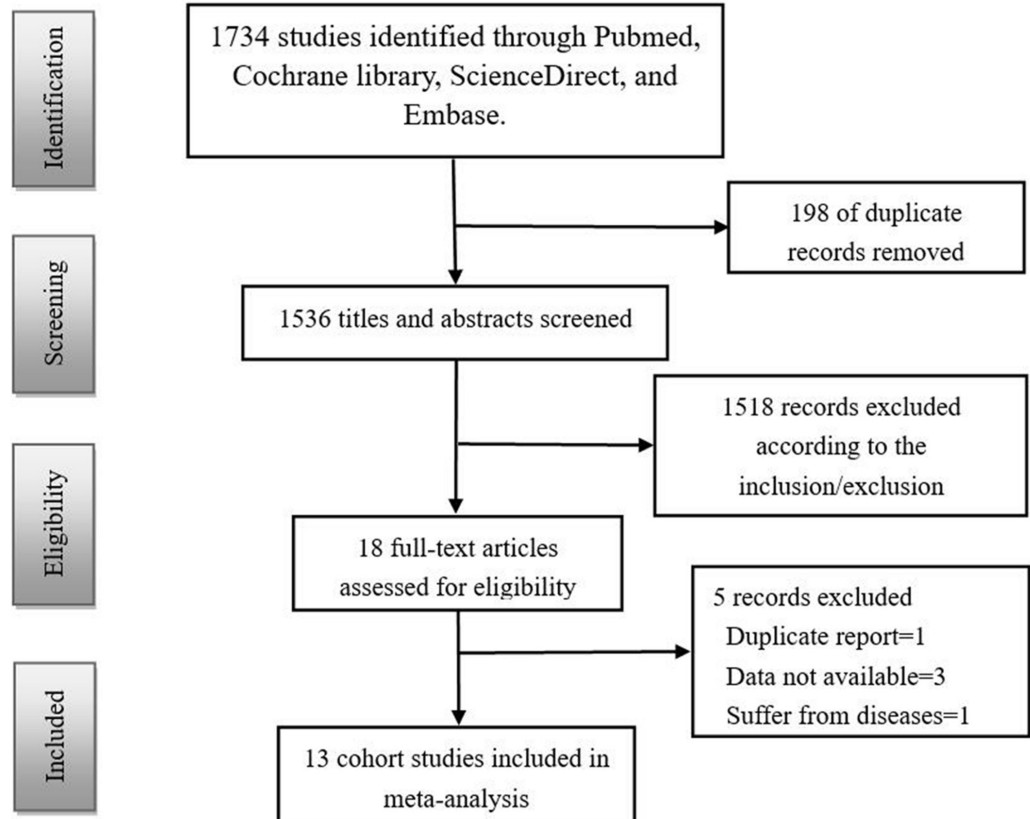

**Figure 1** Flowchart of the literature search and trial selection process.

## RESULTS
### Description of included studies
A total of 1734 studies were identified for initial review using the described search strategies. After removing duplicates, 1536 studies remained. We screened the titles and abstracts of these studies and excluded 1518 records according to the inclusion and exclusion criteria. The 18 remaining full-text articles were then assessed for eligibility. Finally, 13 cohort studies[4 15–18 24–31] were included in the meta-analysis (figure 1), with a total sample of 28 285 pregnant women.

The characteristics and methodological quality of the 13 studies are presented in table 1 and online supplementary table S1. These studies were published between 2010 and 2016; four were conducted in the United states, three in the Netherlands, two in China and one each in Korea, Singapore, Ireland and New Zealand. The average age of the pregnant women in these studies was <30 years for four studies and >30 years for five studies; the average pre-pregnancy BMI of the participants was $<25\,kg/m^2$ in seven studies and $>25\,kg/m^2$ in three studies. Ten studies adjusted for confounders and three studies did not. Five studies collected blood during the first trimester, five during the second trimester, and three during a mixture of the first, second and third trimesters. Five assay methods were used to measure the vitamin D levels of pregnant women, and two criteria were used for the diagnosis of infants who were SGA (birth weight in the lowest 10th or 15th percentile of the reference population). The prevalence of maternal vitamin D deficiency varied from 13.2% to 77.3% (online supplementary table S1). NOS scores were presented as either representing high levels (nine studies) or low levels (four studies) (online supplementary table S2).

### Meta-analysis results
The overall results revealed that maternal vitamin D deficiency during pregnancy was significantly associated with an increased risk of infants who are SGA (pooled OR=1.588; 95% CI 1.138 to 2.216; p<0.01) in the random effects model. A forest plot showing the details is presented in figure 2.

### Subgroup analysis
Due to the existence of heterogeneity ($I^2$=84.2%; p<0.001), subgroup analysis was performed to investigate the possible sources of heterogeneity in the meta-analysis (table 2). The subgroups were created based on cut-off vitamin D levels, measurement of vitamin D, sample size, study quality (NOS score), whether the study adjusted for critical confounders, and the gestational week in which blood sampling was performed. In subgroup analyses, the CIs for each subgroup were overlapped, indicating no significant differences in the effect estimates. Thus, there were no differences in the association between vitamin D deficiency and infants who were SGA based on study quality, time of blood sampling, cut-off vitamin D levels, sample size, adjustment for critical confounders,

**Table 1** Characteristics of the included studies in the present meta-analysis

| Author | Region | Year | Age at baseline (mean, year) | Pre-pregnancy BMI (mean, kg/m²) | Gestational week of blood sampling | Measurement of vitamin D | SGA criteria | Cut-off values | Ethnicity group | OR (95% CI) | Adjusted | NOS score | Sample size |
|---|---|---|---|---|---|---|---|---|---|---|---|---|---|
| Leffelaar[15] | The Netherlands | 2010 | NA | NA | 12–14 weeks | enzyme immunoassay | <10th | <15ng/mL | Dutch (60.3%), Surinamese (6.7%), Turkish (4.0%), Moroccan (6.3%), other non-western (14.2%), other western (8.6%) | 1.90 (1.40 to 2.70) | yes | 8 | 3730 |
| Burris[24] | USA | 2012 | 32.5 | 24.8 | 26–28 weeks | CLIA and RIA | <10th | <10ng/mL | White (83.6%), black (16.4%) | 3.17 (1.16 to 8.63) | yes | 7 | 1133 |
| Zhou[25] | China | 2014 | 29.5 | 20.3 | 16–20 weeks | ECLIA | <10th | <20ng/mL | Asian | 2.46 (0.71 to 8.46) | no | 8 | 1923 |
| Choi[26] | Korea | 2015 | 32.0 | 20.2 | first or second or third trimester | LC-MS/MS | <10th | <20ng/mL | Asian | 0.448 (0.149 to 1.351) | yes | 6 | 220 |
| Ong[18] | Singapore | 2016 | 30.5 | 26.1 | 26–28 weeks | LC-MS/MS | <10th | <20ng/mL | Asian | 1.00 (0.56 to 1.79) | yes | 8 | 910 |
| Kiely[27] | Ireland | 2016 | 30.5 | 24.9 | 14–16 weeks | LC-MS/MS | <10th | <20ng/mL | White (98%), others (2%) | 0.88 (0.60 to 1.28) | yes | 6 | 1768 |
| Scholl[28] | USA | 2014 | 22.8 | 26 | 13.8±5.6 weeks | HPLC | <10th | <20ng/mL | Hispanic (51.4%), non-Hispanic black (34.4%), non-Hispanic white (14.2%) | 0.930 (0.568 to 1.523) | no | 8 | 1045 |
| Chen[4] | China | 2015 | 27.5 | NA | first or second or third trimester | RIA | <10th | <20ng/mL | Asian | 6.47 (4.30 to 9.75) | yes | 6 | 3658 |
| Boyle[29] | New Zealand | 2016 | 30.3 | 24.8 | 15 weeks | LC-MS/MS | <10th | <20ng/mL | NZ European (83.8%), other ethnicities (16.2%) | 1.33 (0.91 to 1.96) | yes | 7 | 2065 |
| Berg[30] | The Netherlands | 2013 | NA | NA | 12.9 weeks | enzyme immunoassay | <10th | <20ng/mL | NA | 1.57 (1.03 to 2.39) | yes | 7 | 2274 |
| Gerand[16] | USA | 2013 | NA | 22.3 | 20.6 weeks | LC-MS/MS | <10th | <15ng/mL | White (52.1%), Black (41.6%), Puerto Rican (6.3%) | 1.284 (1.026 to 1.608) | no | 6 | 2146 |
| Miliku[31] | The Netherlands | 2016 | 29.7 | 23.7 | 20.3 weeks | LC-MS/MS | <15th | <10ng/mL | European (57.3%), Cape Verdean (4.4%), Dutch Antillean (3.5%), Moroccan (6.6%), Surinamese (9.1%), Turkish (9.2%), other (9.9%) | 2.07 (1.33 to 3.22) | yes | 7 | 7176 |
| Nobles[17] | USA | 2015 | NA | >25 | first or second or third trimester | ECLIA | <10th | <20ng/mL | White (75.6%), black (13.5%) | 2.14 (0.67 to 6.88) | yes | 8 | 237 |

CI, confidence interval; CLIA, chemiluminescence immunoassay; ECLIA, electrochemiluminescence immunoassay; HPLC, high-performance liquid chromatography; LC-MS/MS, liquid chromatography tandem mass spectrometry; NA, not available; OR, odds ratio; RIA, radioimmunoassay; SGA, small for gestational age.

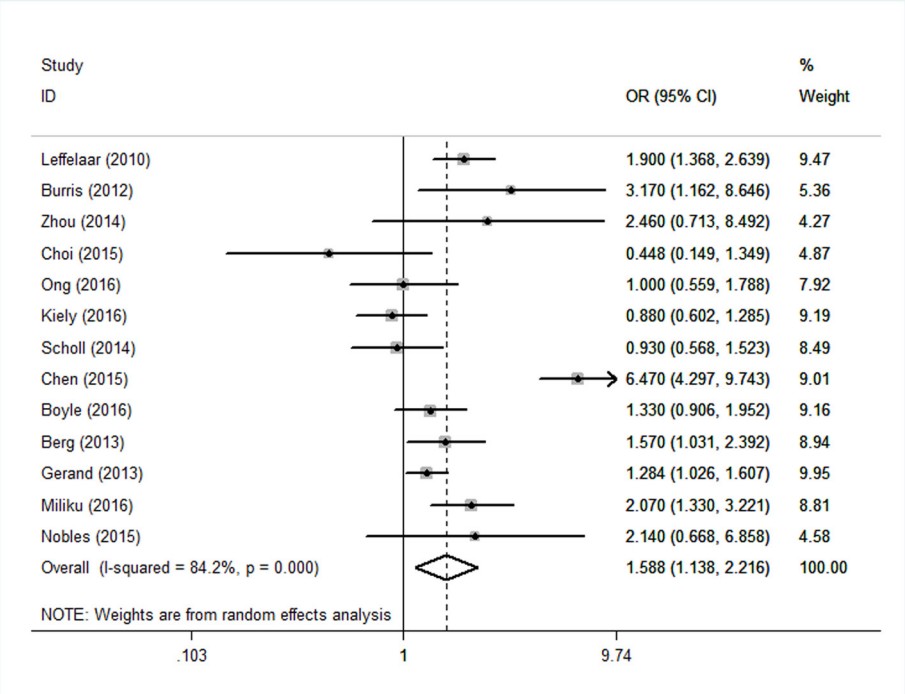

**Figure 2** Forest plots of summary crude odds ratios of the association between vitamin D deficiency.

**Table 2** Subgroup analysis of the association between maternal vitamin D deficiency and SGA

| Stratification group | N | p Value for OR | OR (95% CI) | Heterogeneity test $I^2$(%) | p Value |
|---|---|---|---|---|---|
| **Study quality (NOS)** | | | | | |
| High | 9[15 17 18 24 25 28–31] | <0.001 | 1.555 (1.239 to 1.951) | 37.6 | 0.118 |
| Low | 4[4 16 26 27] | 0.440 | 1.441 (0.570 to 3.641) | 95.2 | <0.001 |
| **Gestation of blood sampling** | | | | | |
| first trimester | 5[15 27–30] | 0.104 | 1.286 (0.950 to 1.741) | 65.9 | 0.020 |
| second trimester | 5[16 18 24 25 31] | 0.011 | 1.577 (1.110 to 2.240) | 51.1 | 0.085 |
| mixed (first or second or third) | 3[4 17 26] | 0.432 | | 90.6 | <0.001 |
| **Cut-off values** | | | | | |
| <10 ng/mL | 2[24 31] | 0.001 | 2.219 (1.480 to 3.325) | 0 | 0.446 |
| <15 ng/mL | 2[15 16] | 0.029 | 1.532 (1.046 to 2.246) | 73.2 | 0.054 |
| <20 ng/mL | 9[4 17 18 25–30] | 0.172 | 1.448 (0.851 to 2.465) | 88.2 | <0.001 |
| **Sample size** | | | | | |
| >1000 | 10[4 15 16 24 25 27–31] | 0.003 | 1.760 (1.217 to 2.544) | 86.8 | <0.001 |
| >1000 | 3[17 18 26] | 0.946 | 0.975 (0.476 to 1.999) | 45.5 | 0.160 |
| **Adjust for critical confounders** | | | | | |
| yes | 10[4 15 17 18 24 26 29–31] | 0.018 | 1.681 (1.094 to 2.584) | 86.3 | <0.001 |
| no | 3[16 25 28] | 0.180 | 1.219 (0.912 to 1.629) | 22.3 | 0.276 |
| **Measurement of vitamin D** | | | | | |
| LC-MS/MS | 6[16 18 26 27 29 31] | 0.204 | 1.195 (0.908 to 1.573) | 59.5 | 0.031 |
| Others | 7[4 15 17 24 25 28 29] | 0.006 | 2.224 (1.263 to 3.918) | 85.8 | <0.001 |

**Table 3** Sensitivity analyses of the association between vitamin D deficiency and SGA

| Study omitted | OR (95% CI) | p value | I² (%) | p value |
|---|---|---|---|---|
| Leffelaar[15] | 1.559 (1.074 to 2.263) | 0.020 | 85.2 | <0.001 |
| Burris[24] | 1.527 (1.084 to 2.152) | 0.016 | 85.1 | <0.001 |
| Zhou[25] | 1.557 (1.105 to 2.195) | 0.011 | 85.4 | <0.001 |
| Choi[26] | 1.693 (1.211 to 2.366) | 0.002 | 84.5 | <0.001 |
| Ong[18] | 1.652 (1.162 to 2.350) | 0.005 | 85.0 | <0.001 |
| Kiely[27] | 1.686 (1.191 to 2.387) | 0.003 | 83.4 | <0.001 |
| Scholl[28] | 1.669 (1.174 to 2.371) | 0.004 | 84.6 | <0.001 |
| Chen[4] | 1.366 (1.103 to 1.692) | 0.004 | 55.4 | 0.010 |
| Boyle[29] | 1.616 (1.118 to 2.335) | 0.011 | 85.4 | <0.001 |
| Berg[30] | 1.590 (1.102 to 2.293) | 0.013 | 85.4 | <0.001 |
| Gerand[16] | 1.624 (1.100 to 2.397) | 0.015 | 84.7 | <0.001 |
| Miliku[31] | 1.548 (1.079 to 2.220) | 0.018 | 85.1 | <0.001 |
| Nobles[17] | 1.565 (1.109 to 2.209) | 0.011 | 85.4 | <0.001 |

and measurement of vitamin D (table 2). However, we did not conduct subgroup analyses regarding ethnicity, pre-pregnancy BMI, gestational age of infant at birth, and season during which blood sampling was performed due to insufficient or unspecific data in some studies.

### Sensitivity analysis and publication bias

To evaluate the stability of our results, sensitivity analysis was performed. Chen's study[4] was discovered to be responsible for most of the heterogeneity in this meta-analysis. Excluding that study resulted in low heterogeneity among the remaining studies (I²=55.4%, p=0.010) with a pooled OR of 1.336 (95% CI 1.103 to 1.692). Furthermore, there were no obvious changes in the pooled ORs as a result of

the exclusion of any other single study; the pooled ORs obtained ranged from 1.366 (95% CI 1.103 to 1.692) to 1.693 (95% CI 1.211 to 2.366), and each was statistically significant (table 3). Additionally, no publication bias was identified using Begg's test (p=0.669) and Egger's regression test (p=0.815). A funnel plot displaying the details is presented in figure 3.

### DISCUSSION

The prevalence of vitamin D deficiency during pregnancy and its association with the risk of infants who are SGA is attracting increasing attention. The present meta-analysis

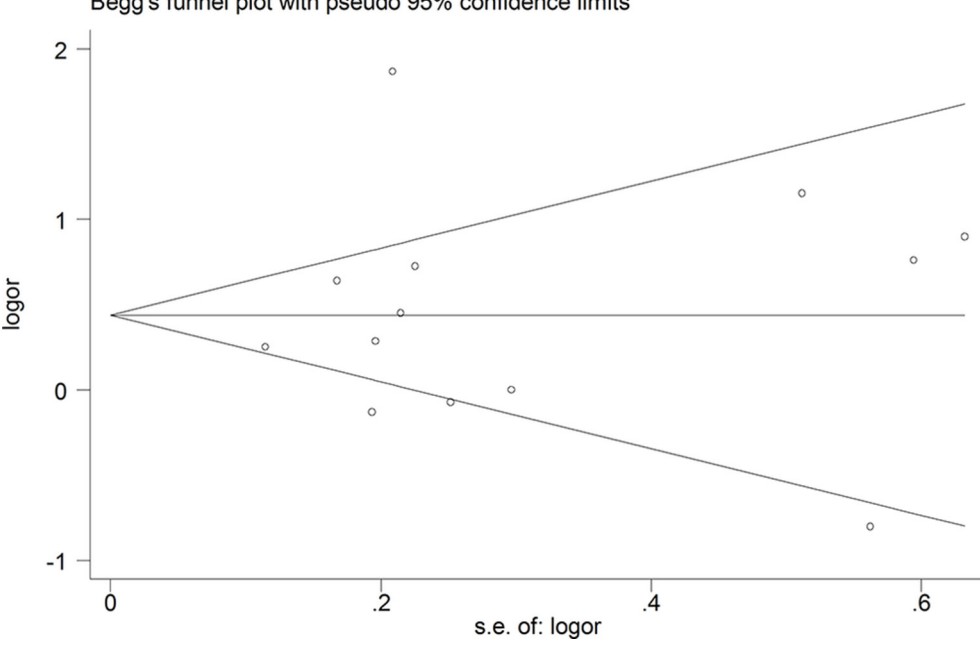

**Figure 3** Funnel plot for small for gestational age. Log odds ratio (OR) of the individual studies plotted against the SE of log OR.

of prospective cohort studies suggested that vitamin D deficiency is significantly associated with a higher risk of SGA. No publication bias was detected, and sensitivity analysis demonstrated that no single study markedly affected the results, which indicated that the results of our meta-analysis are stable and reliable.

The findings of our study are in agreement with several previous studies. One previous meta-analysis showed that low maternal vitamin D levels during pregnancy may be associated with an increased risk of SGA, gestational diabetes mellitus and preterm birth.[5] Similarly, another vital meta-analysis suggested that vitamin D insufficiency was associated with an increased risk of SGA, preeclampsia and bacterial vaginosis.[6] However, those meta-analyses included both case-control and prospective cohort studies and did not include the most recently published cohort studies; additionally, they did not evaluate the association using specific subgroup analysis. Moreover, the cut-off vitamin D levels differed between different studies. Thus, we conducted this meta-analysis to provide stronger evidence for the association between vitamin D and SGA.

The heterogeneity test (Cochran Q test) revealed significant heterogeneity among the studies in this meta-analysis. We investigated the potential factors affecting the results by performing subgroup analysis. The results of the subgroup analyses demonstrated no significant differences in the association between vitamin D deficiency and SGA based on study quality, gestational week during which blood sampling was performed, cut-off values, sample size, adjustment for critical confounders and measurement of vitamin D; however, other factors may have contributed to the heterogeneity in our meta-analysis. Maternal ethnicity, season during which blood sampling was performed, and sunlight exposure and diet during pregnancy are confounding factors for the association between vitamin D deficiency and SGA. Sensitivity analysis revealed that exclusion of any single study did not materially alter the overall combined effect, but also that Chen's study[4] probably contributed greatly to the heterogeneity observed. Therefore, we should interpret the results of this meta-analysis objectively.

The underlying mechanism through which vitamin D deficiency increases the risk of SGA infants is not entirely clear but may be related to the inflammatory response. Vitamin D deficiency can increase levels of proinflammatory cytokines, leading to oxidative stress. Lower 25(OH)D status is associated with increased vascular endothelial cell expression of nuclear factor κB (NFκB) and interleukin 6 and with decreased expression of vitamin D receptor and 1-α hydroxylase.[32] One study reported that levels of proinflammatory cytokines in the cord blood of infants who were SGA were significantly higher than those in the cord blood of infants who were not SGA.[33] Mullins et al[34] reported that more tumour necrosis factor (TNF-α) was expressed in pregnant women with infants who were SGA than in those with infants who were not SGA. As a critical inflammatory factor, TNF-α was previously revealed to inhibit placental hormone synthesis and stimulate calcitriol catabolism through the regulation of enzymes.[35] Vitamin D may also play a crucial role in innate and adaptive immunity by inhibiting the decidual NFκB pathway to reduce inflammatory response, because NFκB is a main transcription factor of inflammatory mediators.[36]

Maternal vitamin D deficiency is common and is influenced by numerous variables, including ethnicity, region of residence, skin pigmentation, sun exposure, season, age and vitamin D supplementation.[37] The American Association of Endocrinology states that pregnant women require at least 600 IU/day of vitamin D and that at least 1500–2000 IU/day of vitamin D may be necessary to maintain a blood level of >30 ng/mL.[38] However, recommendations for vitamin D supplementation in pregnant women are scant. Vitamin D supplementation during pregnancy was suggested as an intervention to prevent adverse pregnancy outcomes.[39] A randomised controlled trial reported that maternal vitamin D supplementation of 2000 or 4000 IU/day appeared to be safe during pregnancy, and the most effective supplementation for optimising serum vitamin D concentrations in mothers and their infants was 4000 IU/day.[40] This result is consistent with another randomised controlled trial in Pakistan.[41] In two studies, low vitamin D levels during pregnancy increased the risk of SGA, however vitamin D supplementation did not significantly reduce the risk of SGA (OR=0.78, 95% CI 0.50 to 1.21[42] and OR=0.67, 95% CI 0.40 to 1.11)[43]. Another study found it difficult to draw a final conclusion regarding the need for vitamin D supplementation during pregnancy.[44] Therefore, larger randomised controlled trials are required to assess the value of such interventions, and will have a significant impact on the guidance regarding perinatal care.

Our study had several strengths. First, to ensure that evidence was reliable, we included only prospective cohort studies, which have more advantages than case-control studies. Second, no publication bias was present in our meta-analysis, indicating that its results may be unbiased and credible. Finally, our study's subgroup analysis enabled thorough understanding of the current evidence. However, several limitations should also be acknowledged. The association between maternal vitamin D status and SGA risk may have been affected by confounding factors such as pre-pregnancy BMI, age, education, ethnicity and sunlight exposure; not all the included studies controlled for these confounding factors. Additionally, the included studies had different definitions of vitamin D deficiency, insufficiency or sufficiency, which may have affected the results. Lastly, pooled data without detailed individual information were used to perform the meta-analysis, which restricted us from obtaining comprehensive results.

## CONCLUSIONS

The present study indicates that low vitamin D levels in pregnant women are associated with an increased risk of infants who are SGA. Further confirmation of these findings in larger-sample studies is required. The role of vitamin D in the pathogenesis of SGA should be emphasised. Additionally, early screening for vitamin D deficiency among pregnant women may be necessary.

**Contributors** FT contributed to study design; SL, XW and BZ contributed analysis tools and methods; YC analysed the data and drafted the manuscript; BZ and FT revised the manuscript. All authors read and approved the final version of the manuscript.

**Funding** The authors are grateful for the financial support offered by the National Natural Science Foundation of China (81330068, 81573168).

**Competing interests** None declared.

**Patient consent** Obtained.

**Provenance and peer review** Not commissioned; externally peer reviewed.

**Data sharing statement** No additional data are available.

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
