## [Reviewer comments · BMJ Open]

ARTICLE DETAILS

TITLE (PROVISIONAL)	Association between maternal vitamin D deficiency and small for gestational age: evidence from a meta-analysis of prospective cohort studies
AUTHORS	Chen, Yao; Zhu, Beibei; Wu, Xiaoyan; Li, Si; Tao, Fangbiao

VERSION 1 - REVIEW

REVIEWER	Shu Qin Wei, Wei Guang Bi University of Montréal, Canada
REVIEW RETURNED	12-Mar-2017

GENERAL COMMENTS	The authors conducted a systematic review and meta-analysis of the relationship between maternal vitamin D deficiency and small for gestational age from prospective cohort studies. They concluded that vitamin D deficiency is associated with an increased risk of SGA. The conclusions are reasonable and well grounded in the data presented. However, there are some concerns should be addressed. The abstract results section needs absolute numbers (percentages or actual numbers) that are helpful for physicians when counseling patients, instead of just odds ratios and confidence intervals. 'Maternal with circulating 25-hydroxyvitamin D [25(OH)D] deficiency experienced had an increased risk of SGA' need rephrase. 'When stratified according to blood sampling weeks', it may be stratified by trimester? As some studies were mixed with first, second or third trimester, how did the authors stratify these studies? Introduction: the sentence at second paragraph: 'it might also do a lot harm to other well-beings throughout childhood to adulthood' need rephrase. 'Vitamin D deficiency' was not defined in the manuscript. Table 1 should provide the information on ethnicity and season. Table 2 should with references for each subgroup. For example: Caucasian were from 9 studies, which were they? As well, it seems only mentioned Caucasian and Asian, it seems had more ethnicities in some studies. For example: The study Gernand 2013 (ref15) included white, black and Puerto Rican. How did the ethnicity subgroup analysed? Table 3 it is not clearly how did the sensitivity analyses conduct? Why need have individual study OR in the table? It should be in the meta-analysis. It might be worthwhile to conduct a sensitivity analysis for the measurement of LC-MS/MS, which is best method for measurement of vitamin D levels.
--

REVIEWER	Barbara Willey London School of Hygiene & Tropical Medicine
REVIEW RETURNED	04-Apr-2017

GENERAL COMMENTS	Association between maternal vitamin D deficiency and small for gestational age: evidence from a meta-analysis of prospective cohort studies Thank you for this interesting systematic review and meta analysis investigating the association between maternal vitamin D deficiency and SGA, based on cohort evidence. Please find below specific comments. Abstract 1. Please see further detail below under methods (statistical analysis) in order to update interpretation of sub group analyses in the abstract. Introduction 2. Statement on association between SGA and mortality- this has been quantified in a pooled analysis and estimated at 1.8 for neonatal mortality and 1.9 for post neonatal mortality (reference: Katz et al. Mortality risk in preterm and small-for-gestational-age infants in low-income and middle-income countries: a pooled country analysis. The Lancet 2013). 3. The phrase in second paragraph of introduction beginning “Even worse” perhaps needs rephrasing. It suggests that influences of SGA across the life course are worse than SGA impact on mortality, which I don’t think this is what author’s mean to suggest. Methods 4. Search strategy:  • Please specify whether these were free text searches or whether subject headings were used e.g. MeSH terms in PubMed? • A considerably larger volume of papers were found in the science direct database than in others, please provide details of the search strategy and how this was translated to each database. 5. Selection criteria:  • Please provide clarification on the selection/ inclusion/ exclusion criteria for studies in this systematic review. Using the PICOS format for presenting these would be useful. • In particular, it was not clear whether the systematic review had as an objective to quantify the size of the association between vitamin D deficiency and SGA, or whether supplementation with Vitamin D was also of interest. • Further details on the exposure and outcome are also necessary e.g. was vitamin D deficiency always defined as a binary variable, what was the definition of SGA used in the studies, was this always <10th percentile, was the same
--

reference population used across all studies?

- Provide rationale for the study design inclusion criteria (restriction to cohort studies) in this section.
- Figure 1 suggests “suffer from diseases” as an exclusion criteria, so this should also be specified in the text.

6. Under data extraction:

- Authors do not mention that country and ethnicity are extracted. Region is presented in Table 1, but it is important to highlight ethnicity within this region, e.g. cohorts including various ethnicities. It is not necessarily correct that region conducted (country) is the same as ethnic group, as suggested in results section (9 caucasian, 4 asian).
- It would also be interesting to present study summary measure of education status of the mother or other socio-economic indicators.
- It is also important to present study summary measure of gestational age of the infant at birth. These are both important factors associated with SGA.
- It would also be interesting to present information on latitude and time of year data collected etc. (in terms of sunlight).

7. Statistical Analysis:

- Fixed vs random effects: the authors have carried out a random effects meta analysis, which seems appropriate because the rationale that the studies share a common effect size doesn't seem appropriate. However, the reasons the authors use to justify a random effects model is inappropriate. It is not appropriate to use a % I² or p value cut off of the I² to determine random vs fixed effects modelling, and in fact the Cochrane Handbook explicitly discourages this, and this is not what Higgins et al suggest in their paper (ref #20). Could authors please revise this section on justification for random effects in line with recommended best practice for meta analyses (e.g. see Borenstein et al Res. Syn. Meth. 2010, 1 97—111).
- Sub group analysis: authors carry out six sub group analyses. Please clarify whether these were pre-specified, and provide a clear scientific rationale for each. Sub group analyses are often used to explore heterogeneity, and authors need to guide readers further on their choice, as well as providing clear justification for the large number of subgroups. This issue is subject to multiple testing (e.g. see Bender J Clin Epidemiol 61:857-865.). Also consider that throughout these subgroup analyses there are very few studies in some of the categories (e.g. 2, 3, 4 studies only). If subgroups are appropriately justified, it is appropriate to examine whether confidence intervals overlap for binary categories or use a test for interaction (e.g. see Borenstein 2008, Introduction to Meta-analysis) for multiple categories, rather than relying on the individual subgroup level p values for interpretation (see Cochrane handbook).

Results

8. Please provide a more detailed summary/ overview of the

quality of the studies included beyond reporting the Newcastle Ottawa tool score.

9. Please adjust interpretation of results in the light of above comments on subgroup analysis. In particular, findings are reported with undue reliance on p values for interpretation, which is inappropriate, in fact emphasis should be placed on the magnitude of summary statistics and the overlap of CIs, as well as considering the interpretation for public health and clinical medicine for the different subgroups- if different behaviours wouldn't be carried out for the subgroups it may not be appropriate to present that subgroup pooled estimate, but rather stick with the overall pooled estimate.

Discussion

10. Comment on results differing between Caucasian and African women is not appropriately supported given the current classification of country of study rather than ethnic group composition of cohort studies included.
11. Authors suggest larger RCTs of vitamin D supplementation are required, although the previous reference cites a systematic review and meta analysis that concluded supplementation did not have a significant impact in reducing SGA. Please support the argument for further or larger RCTs more fully.

Tables

12. Table 1: please specify whether age and BMI measures are means.
13. Table 1: please include ethnic group and socio-economic status, mother's education, gestational age at delivery where available
14. Table 1: please include the summary statistic (OR and 95% CIs) for each study in this table.
15. Table 3: I think it would be helpful for authors to draw attention to the result in the sensitivity analysis (table 3) that shows that exclusion of the Chen 2015 study reduces the pooled odds ratio by >10% in comparison to the overall pooled effect of 1.57.

Figures

16. Figure 1: the labelling could have been clearer e.g. 1536 titles and abstracts screened: PRISMA guidelines suggest standard wording. The numbers of papers quoted in the text (n=1537) doesn't follow the detail in figure 1.
17. Figure 2 forest plot: The confidence intervals for Zhou 2014 are very wide, which is surprising given the reasonable sample size of ~1900 vs the other study sizes included in this meta analysis. I am not sure where the data comes from as a very cursory examination of the paper and results for the association with SGA shown in table 4 suggest an odds ratio of 0.98. Clarification would be helpful.

VERSION 1 – AUTHOR RESPONSE

Responses to reviewers

Reviewer # 1:

Comment 1: The abstract results section needs absolute numbers (percentages or actual numbers) that are helpful for physicians when counseling patients, instead of just odds ratios and confidence intervals. 'Maternal with circulating 25-hydroxyvitamin D [25(OH)D] deficiency experienced had an increased risk of SGA' need rephrase. 'When stratified according to blood sampling weeks', it may be stratified by trimester? As some studies were mixed with first, second or third trimester, how did the authors stratify these studies?

Response:

We appreciate you for this point. First, according to the comment from the reviewer, we have added some absolute numbers (percentages or actual numbers) in our abstract as much as possible. However, our study was a merger OR value that we could not get more information or data. See P.02, Line 27-31. Second, we have rephrased the sentence (Maternal with circulating 25-hydroxyvitamin D [25(OH)D] deficiency experienced had an increased risk of SGA.) as follows: "Pooled overall ORs for babies with SGA were 1.588 (95% CI 1.138 to 2.216; P<0.01) for women whose vitamin D deficiency." See P.02, Line 28-30. Third, we revised the results of subgroup analysis according to whether confidence intervals overlap. The results showed that "there were no significant differences in the association between vitamin D deficiency and SGA based on study quality, gestation of blood sampling, cut-off values, sample size, adjust for critical confounders and measurement of vitamin D." See P.02 Line 31-34. According to the data of blood sampling weeks was extracted from original study, however, some studies provide specific weeks of gestation, and we artificially divide these gestational weeks into first or second trimester or mixed. Thus, we only collected three kinds of information about blood gestational age (first trimester/ second trimester/ first or second or third trimester) at last, the information of late pregnancy was not collected separately. According to the reviewer's comments, we have added the information of mixed trimester in table 2. See Table 2
Comment 2: Introduction: the sentence at second paragraph: 'it might also do a lot harm to other well-beings throughout childhood to adulthood' need rephrase.

Response:

Thank you very much for the suggestions. We agree with the comment and re-wrote the sentence in the revised manuscript as follows:

"In addition, it might be strongly related to adverse health outcomes in adult life." See P. 04, Line 69-70.

Comment 3: 'Vitamin D deficiency' was not defined in the manuscript.

Response:

Thanks for your concern. We have added the definition of 'vitamin D deficiency' in the selection criteria section as follows:

"3) the outcome was SGA and the control group included maternal without SGA, the exposure was 'vitamin D deficiency' [25(OH)D<20ng/ml]" See P.05, Line 102-104.

Comment 4: Table 1 should provide the information on ethnicity and season.

Response:

Thanks for your comment very much. We have added the information on ethnicity in table 1 and added the information on season in supplementary table S1. See Table1 and supplementary Table S1.

Comment 5: Table 2 should with references for each subgroup. For example: Caucasian were from 9 studies, which were they? As well, it seems only mentioned Caucasian and Asian, it seems had more ethnicities in some studies. For example: The study Gernand 2013 (ref15) included white, black and Puerto Rican. How did the ethnicity subgroup analyses?

Response:

Thanks for your further suggestion to help us improving our manuscript. First, relevant references for each subgroup were tagged in table2. See Table 2. Second, we conducted mistakenly country group as the ethnicity group previously, it is not necessarily correct that country (region) is the same as

ethnicity group, thus we have revised the results of this part in subgroup analysis section as follows: "However, we did not conduct subgroup analysis of ethnicity, pre-pregnancy BMI, gestational age of infant at birth and season of blood sample due to insufficient/ unspecific data in some studies." See P.09, Line 204-207

Comment 6: Table 3 it is not clearly how did the sensitivity analyses conduct? Why need have individual study OR in the table? It should be in the meta-analysis.

Response:

Thanks for your comment. To evaluate the stability of our results, sensitivity analysis was performed by the leave one out at a time and checking the consistency of the overall effect estimate. Thus, we calculated the pooled OR for the remaining studies after excluding one study. The OR in Table 3 was pooled OR for remaining studies. See table 3

Comment 7: It might be worthwhile to conduct a sensitivity analysis for the measurement of LC-MS/MS, which is best method for measurement of vitamin D levels.

Response:

Thanks for your comment, we added a subgroup analysis for the measurement of LC-MS/MS in the revised manuscript. See Table 2.

Reviewer # 2:

Comment 1: Please see further detail below under methods (statistical analysis) in order to update interpretation of sub group analyses in the abstract.

Response:

Thank you very much for the suggestions. We have rewritten the abstract in order to update interpretation of sub group analysis as follows:

"Totally 13 cohort studies were included in this meta-analysis containing 28285 individuals from 7 countries. Pooled overall ORs for babies with SGA were 1.588 (95% CI 1.138 to 2.216; $P < 0.01$) for women with vitamin D deficiency. In addition, the prevalence of vitamin D deficiency during pregnancy varied from 13.2% to 77.3%. Subgroup analyses showed that there were no significant differences in the association between vitamin D deficiency and SGA based on study quality, gestation of blood sampling, cut-off values, sample size, adjust for critical confounders and measurement of vitamin D." See P.02, Line 27-34.

Comment 2: Statement on association between SGA and mortality- this has been quantified in a pooled analysis and estimated at 1.8 for neonatal mortality and 1.9 for post neonatal mortality (reference: Katz et al. Mortality risk in preterm and small-for-gestational-age infants in low-income and middle-income countries: a pooled country analysis. The Lancet 2013).

Response:

Thanks for your helpful suggestion. We have added the information of this paper in introduction section as follows:

"Katz J et al.¹¹ have showed that pooled RRs for infants who were SGA were 1.83 for neonatal mortality and 1.90 for post-neonatal morbidity." See P.04, Line 67-69.

Comment 3: The phrase in second paragraph of introduction beginning "Even worse" perhaps needs rephrasing. It suggests that influences of SGA across the life course are worse than SGA impact on mortality, which I don't think this is what author's mean to suggest.

Response:

Thank you very much for the suggestions. We agree with the comment and re-wrote the sentence in the revised manuscript as the following:

"In addition, it might be strongly related to adverse health outcomes in adult life." See P. 04, Line 69-70.

Comment 4: Please specify whether these were free text searches or whether subject headings were used e.g. MeSH terms in PubMed? A considerably larger volume of papers were found in the science direct database than in others, please provide details of the search strategy and how this was translated to each database.

Response:

Thank you for the concern. First, we have specified that we searched those text by using key words

(cholecalciferol, vitamin D, 25-hydroxyvitamin D and small for gestational age were MeSH terms, 25(OH)D, SGA, small-for-gestation age, small size for gestational age were not MeSH terms). See P.05, Line 91-94. Second, the details of search strategy that we have provided in supplementary box S1. See supplementary box S1.

Comment 5: 1) Please provide clarification on the selection/ inclusion/ exclusion criteria for studies in this systematic review. Using the PICOS format for presenting these would be useful. 2) In particular, it was not clear whether the systematic review had as an objective to quantify the size of the association between vitamin D deficiency and SGA, or whether supplementation with Vitamin D was also of interest. 3) Further details on the exposure and outcome are also necessary e.g. was vitamin D deficiency always defined as a binary variable, what was the definition of SGA used in the studies, was this always <10th percentile, was the same reference population used across all studies? 4) Provide rationale for the study design inclusion criteria (restriction to cohort studies) in this section. 5) Figure 1 suggests “suffer from diseases” as an exclusion criteria, so this should also be specified in the text.

Response:

Thanks for your further suggestion to help us improving our manuscript. First, we have re-stored the inclusion criteria for studies in our systematic review as follows:

“1) the population of study was maternal without pre-chronic disease; 2) the study included maternal with singleton gestation; 3) the outcome was SGA and the control group included maternal without SGA, the exposure was ‘vitamin D deficiency’ [25(OH)D<20ng/ml]; 4) studies with data in the form of effect estimate [odds ratio (OR) or risk ratio (RR)] and corresponding 95% confidence interval (CI) or reported data to calculate them; 5) maternal blood samples were taken for assessing 25(OH)D during pregnancy; 6) the study design was cohort studies (to provide more reliable evidence, we only included prospective cohort studies, which have more advantages than case-control studies); 7) published in English.” See P.05, Line 101-110. Second, the focus of our systematic review was to quantify the size of the association between vitamin D deficiency and SGA. At the same time, we were interested in vitamin D supplementation, however, since some researchers have done systematic review about this part, thus we discussed a little for vitamin D supplementation in discussion section. Third, we have added the relevant information of the exposure and outcome in table 1. See Table 1. Fourth, we have given an explanation for restriction to cohort studies in inclusion criteria section as follows:

“6) the study design was cohort studies (to provide more reliable evidence, we only included prospective cohort studies, which have more advantages than case-control studies);” See P.05, Line 108-110. Finally, we also have added the exclusion criteria (suffer from diseases) in inclusion criteria section as follows:

“1) the population of study was pregnant women without pre-chronic disease;” See P.05, Line 101

Comment 6: 1) Authors do not mention that country and ethnicity are extracted. Region is presented in Table 1, but it is important to highlight ethnicity within this region, e.g. cohorts including various ethnicities. It is not necessarily correct that region conducted (country) is the same as ethnic group, as suggested in results section (9 Caucasian, 4 Asian). 2) It would also be interesting to present study summary measure of education status of the mother or other socio-economic indicators. 3) It is also important to present study summary measure of gestational age of the infant at birth. These are both important factors associated with SGA. 4) It would also be interesting to present information on latitude and time of year data collected etc. (in terms of sunlight).

Response:

We appreciate you for this valuable comment. First, We have revised the mistakes about ethnicity, and added the information of country and ethnicity in table 1. See table 1. We do the biggest possible to extract relevant information on education status, gestational age of the infant at birth, latitude and time of year data collected in table 1 and supplementary table S1. See table 1 and supplementary S1. Third, the information on socio-economic was not extracted due to severe deficient data.

Comment 7: 1) Fixed vs random effects: the authors have carried out a random effects meta-analysis, which seems appropriate because the rationale that the studies share a common effect size doesn't

seem appropriate. However, the reasons the authors use to justify a random effects model is inappropriate. It is not appropriate to use a % I² or p value cut off of the I² to determine random vs fixed effects modelling, and in fact the Cochrane Handbook explicitly discourages this, and this is not what Higgins et al suggest in their paper (ref #20). Could authors please revise this section on justification for random effects in line with recommended best practice for meta analyses (e.g. see Borenstein et al Res. Syn. Meth. 2010, 1 97—111). 2) Sub group analysis: authors carry out six sub group analyses. Please clarify whether these were pre-specified, and provide a clear scientific rationale for each. Sub group analyses are often used to explore heterogeneity, and authors need to guide readers further on their choice, as well as providing clear justification for the large number of subgroups. This issue is subject to multiple testing (e.g. see Bender J Clin Epidemiol 61:857-865.). Also consider that throughout these subgroup analyses there are very few studies in some of the categories (e.g. 2, 3, 4 studies only). If subgroups are appropriately justified, it is appropriate to examine whether confidence intervals overlap for binary categories or use a test for interaction (e.g. see Borenstein 2008, Introduction to Meta-analysis) for multiple categories, rather than relying on the individual subgroup level p values for interpretation (see Cochrane handbook).

Response:

Thanks for your further suggestion to help us improving our manuscript. In the revised version of the manuscript. First, we have revised the reasons for using a random effects model in section as follows: "The random-effects model was usually a more plausible when studies were gathered from the published literature. Therefore, the random effects model was used for this meta-analysis." See P.07 Line 145-147.

Second, in order to explore the possible heterogeneity of meta-analysis, we have specified more than six subgroups at advance, and finally determined six subgroups were able to analysis due to the data of them that could be extracted in original study. But we have no exact reason to do these subgroup analysis, our original intention was to explore the possibility sources of heterogeneity existed according to the characteristics of each study. ("subgroup analyses represent an effort to tackle heterogeneity of treatment effects." Sun X. BMJ 2010). We would like to analyze the subgroups which affect the outcome, but we could not extract the complete/ clear/ useful data. Some observational studies have showed that pregnant women serum 25(OH)D levels was associated with gestational week of blood sample. Morley R et al. found that "gestation length was 0.7 wk (95% CI -1.3, -0.1) shorter and knee-heel length was 4.3 mm smaller (-7.3, -1.3) in infants of 27 mothers with low 25(OH)D (<28nmol/L) at 28-32 wk vs. babies whose mothers had higher concentrations" (Morley R, et al. J Clin Endocrinol Metab, 2006, 91: 906-912). The measurement of LC-MS/MS, which is the best method for measurement of vitamin D levels. Different definition of vitamin D deficiency, insufficiency or sufficiency might have influenced the result. Much to we regret, we could not achieve the suggestions of the reviewers. Our subgroup analysis was based on previous studies, with the aim of exploring possible potential factors.

According to your advice, we have examined whether confidence intervals overlap. After examined, the confidence intervals were overlapped for each subgroup, which showed that no statistically significant difference in the effect estimates. We have adjusted the interpretation of results on subgroup analysis in results section. See P.08-09, Line 200-207.

Comment 8: Please provide a more detailed summary/ overview of the quality of the studies included beyond reporting the Newcastle Ottawa tool score.

Response:

Thank you for your critical comments very much, and we have added a detailed summary of the quality of the studies in the revised manuscript. See supplementary box S1 and supplementary Table S1.

Comment 9: Please adjust interpretation of results in the light of above comments on subgroup analysis. In particular, findings are reported with undue reliance on p values for interpretation, which is inappropriate, in fact emphasis should be placed on the magnitude of summary statistics and the overlap of CIs, as well as considering the interpretation for public health and clinical medicine for the different subgroups- if different behaviors wouldn't be carried out for the subgroups it may not be

appropriate to present that subgroup pooled estimate, but rather stick with the overall pooled estimate.

Response:

First of all, we appreciate you for this valuable comment and we have realized the interpretation of the results for subgroup analysis was inappropriate. According to reviewer's comments, we have revised the content of this part as follows:

"In subgroup analyses, the confidence intervals were overlapped for each subgroup, which showed no statistically significant difference in the effect estimates. Thus, there were no differences in the association between vitamin D deficiency with SGA based on study quality, gestation of blood sampling, cut-off values, sample size, adjust for critical confounders and measurement of vitamin D (Table 2). However, we did not conduct subgroup analyses of ethnicity, pre-pregnancy BMI, gestational age of infant at birth and season of blood sample due to insufficient/ unspecific data in some studies." See P.08-09, Line 109-206.

Comment 10: Comment on results differing between Caucasian and African women is not appropriately supported given the current classification of country of study rather than ethnic group composition of cohort studies included.

Response:

Thank you for your comments very much. According to your advice, we have revised the interpretation of ethnicity in discussion section as follows:

"Although the results of subgroup analyses showed that there were no significant differences in the association between vitamin D deficiency with SGA based on study quality, gestation of blood sampling, cut-off values, sample size, adjust for critical confounders and measurement of vitamin D, there may be other potential factors contributing to the heterogeneity in our meta-analysis. The different for ethnicity of the maternal, season of blood sample, sunlight exposure and diet during pregnancy are confounding factors for the association between vitamin D deficiency and SGA." See P.10-11, Line 256-263.

Comment 11: Authors suggest larger RCTs of vitamin D supplementation are required, although the previous reference cites a systematic review and meta-analysis that concluded supplementation did not have a significant impact in reducing SGA. Please support the argument for further or larger RCTs more fully.

Response:

Thank you for your comments very much. We added two papers in the discussion section in order to support the argument for further or larger RCTs more fully. See P.12, Line 304-305

Comment 12: Table 1: please specify whether age and BMI measures are means.

Response:

Thank you very much for the suggestions. We extracted the mean for the age and pre-pregnancy BMI, the annotation was added in table 1. See table 1.

Comment 13: Table 1: please include ethnic group and socio-economic status, mother's education, gestational age at delivery where available.

Response:

Thank you very much for the suggestions. We have added the information of ethnic group in table 1 and we also have added the information of mother's education and gestational age at delivery in supplementary table S1. But socio-economic status was not extracted due to insufficient data in majority studies. See table 1 and supplementary S1.

Comment 14: Table 1: please include the summary statistic (OR and 95%CIs) for each study in this table.

Response:

Thank you very much for the suggestions. We have added the summary statistic (OR and 95%CIs) for each study in table 1. See table 1.

Comment 15: Table 3: I think it would be helpful for authors to draw attention to the result in the sensitivity analysis (table 3) that shows that exclusion of the Chen 2015 study reduces the pooled odds ratio by >10% in comparison to the overall pooled effect of 1.57.

Response:

Thanks for your precious comments. We reinterpreted the results of the sensitivity analysis as following:

“Chen’s study was responsible for most of the heterogeneity in this meta-analysis. Low heterogeneity was observed among the remaining studies ($I^2=55.4\%$, $P=0.010$) and pooled OR was 1.336 (95% CI 1.103 to 1.692) after excluding Chen’s study⁴. Furthermore, there were no obvious changes in the pooled ORs as a result of the exclusion of any other single study. The pooled ORs ranged from 1.366 (95% CI 1.103 to 1.692) to 1.693 (95% CI 1.211 to 2.366), and each was statistically significant”. See P.09, Line 221-227.

Comment 16: Figure 1: the labelling could have been clearer e.g. 1536 titles and abstracts screened: PRISMA guidelines suggest standard wording. The numbers of papers quoted in the txt (n=1537) doesn’t follow the detail in figure 1.

Response:

Thanks for your comment very much. First, we have modified the labelling of figure1. See figure 1. Second, I am sorry for the numbers quoted in the text doesn’t follow the detail in figure 1, the true numbers are 1536. We have removed all typos. See P.07, Line 160.

Comment 17: Figure 2 forest plot: The confidence intervals for Zhou 2014 are very wide, which is surprising given the reasonable sample size of ~1900 vs the other study sizes included in this meta-analysis. I am not sure where the data comes from as a very cursory examination of the paper and results for the association with SGA shown in table 4 suggest an odds ratio of 0.98. Clarification would be helpful.

Response:

We appreciate you for this point. We apologize for the data problems in the original manuscript. I am sorry that I had made a stupid mistake. The value of OR and 95% CI for Zhou (2014) were calculated by using data from observed articles to construct 2x2 tables of low vitamin D status versus the presence or absence of SGA. In fact, the correct value is 2.46 (95%CI 0.71 to 8.46). But I wrote falsely 2.46 (95%CI 0.17 to 8.46) in the meta-analysis, thus, led to the confidence intervals for Zhou 2014 are very wide at the time of analysis. We are sorry for our incorrect writing. In the Zhou 2014 paper, an odds ratio of 0.98 was represented that prevalence of SGA in high level group of maternal vitamin D (≥ 30 ng/ml) compared to low and medium level groups (as reference), which was not the results we want. We have removed all typos. See Table 1 and Figure 2.

Once again, special thanks to you for your good comments.

VERSION 2 – REVIEW

REVIEWER	Barbara Willey London School of Hygiene and Tropical Medicine London UK
REVIEW RETURNED	06-Jun-2017

GENERAL COMMENTS	Revised version of paper- addressed all my comments fully. Recommend acceptance.
---